# The prevalence of urogenital and intestinal s*chistosomiasis* among school age children (6–13 years) in the Okavango Delta in Botswana

**Nthabiseng A. Phaladze**[1]*, **Lebotse Molefi**[2], **Olekae T. Thakadu**[3], **Onalenna Tsima**[1], **Barbara N. Ngwenya**[3], **Tuduetso L. Molefi**[4], **Wananani B. Tshiamo**[1]

**1** School of Nursing, University of Botswana, Gaborone, Botswana, **2** School of Allied Health Professions, Gaborone, Botswana, **3** Okavango Research Institute, University of Botswana, Gaborone, Botswana, **4** Neglected Tropical Diseases Program, Botswana Ministry of Health, Gaborone, Botswana

* phaladze@ub.ac.bw, nphaladze@yahoo.co.uk

**Data Availability Statement:** All relevant data are within the paper and its Supporting Information files.

## Abstract

This study sought to investigate prevalence of urogenital and intestinal schistosomiasis among school age children 6–13 years in selected communities in the Okavango Delta. The termination of the Botswana national schistosomiasis control program in 1993 contributed to its neglect. An outbreak of schistosomiasis in 2017 at one of the primary schools in the northeastern part of the country resulted in 42 positive cases, indicating that the disease exists. A total of 1,611 school age children 6–13 years were randomly selected from school registers in 10 primary schools; from which 1603 urine and 1404 stool samples were collected. Macroscopic examination of urine and stool for color, odor, blood; viscosity, consistency, and the presence of worms. Urine filtration and centrifugation methods were used to increase sensitivity of detecting parasite ova. Kato-Katz and Formalin-Ether were used for the examination of stool samples. Data were analyzed using SPSS version 25. Results were expressed as odds ratio (*OR*) with their 95% CI and statistical significance set at p < 0.05. A total of (n = 1611) school age children 6–13 years participated in the study, mean age 9.7years (SD 2.06), females (54%) and males (46%). Results indicated an overall prevalence of *SS. hematobium* and *S.mansoni* at 8.7% and 0.64% respectively. Intensity of *SS. hematobium* was generally light (97.6%) and heavy intensity (2.4%). Results also revealed a knowledge deficit, about 58% of children had never heard of bilharzia even though they lived in communities where the disease was previously endemic. Learners who had a family member who previously suffered from schistosomiasis had higher knowledge than those who did not. Interestingly, these learners were likely to engage in risky behaviors compared to those with lower knowledge of the disease. An integrated approach that emphasizes health education, mass drug administration, water, sanitation, and hygiene infrastructure should be prioritized for prevention and control of schistosomiasis.

**Funding:** Funding information has been removed from the Acknowledgments section. We would like to maintain the Funding Statement as it reads "This study was funded by the National Institute of Health Research (Grant Number: 16/136/33 NIHR). The funders had no role in study design, data collection and analysis, decision to publish, or preparation of the manuscript." Any additional information provided contrary to the original statement should be deleted.

**Competing interests:** I declare that the authors have no conflict of interest. Mr . Lebotse Molefi is a paid employee of Quality Anchor Consultants PTY. LTD, but was not affiliated with the company at the time this study was conducted. There are no patents, products in development or marketed products associated with this research to declare. This does not alter our adherence to PLOS ONE policies on sharing data and materials.

## 1. Introduction

Schistosomiasis is considered the second most neglected tropical disease (NTD) after hookworm. It mainly affects developing countries where water resources and poor sanitation allow development and infection of snails respectively [1]. Schistosomiasis affects a large proportion of children under 14 years of age, including at least 25 million preschool aged going children [2]. Schistosomiasis is a parasitic disease caused by blood vessel-dwelling flukes of the genus s*chistosoma*. There are several species in the genus but primarily *S. hematobium* which causes urinary schistosomiasis, *S. mansoni* and *S. japonicum* which both cause intestinal schistosomiasis in humans [3]. are the most prevalent. The invading parasites in Botswana are *S. mansoni* and *S. haematobium*, *with* prevalence rates fluctuating depending on rainfall. Prevalence of schistosomiasis in Botswana has been as high as 80% or as low as 20% in defined geographical areas [4].

The effects of schistosomiasis among children are diverse and alarming, such as urethral and bladder fibrosis, hydronephrosis, hepatosplenomegaly, and can progress into bladder cancer and colorectal cancer as a possible late-stage complication [5, 6]. found that schistosomiasis has profound negative effects on child development, outcome of pregnancy, and agricultural productivity, thus a key reason why the most inhabitants of sub-Saharan Africa continue to live in poverty. Schistosomiasis causes inflammation of the mucosa, hyperplasia of cells and gut ulceration [7], which in turn decreases food intake and nutrient utilization within the host body [8–10], all of which contribute to childhood stunting [11]. The disease can lead to chronic ill health and is considered a major public health concern mostly in rural dwellers of tropical and sub-tropical regions of the world [12].

Human contact with snail habitats is crucial to the establishment of transmission and children are more frequently and more severely infected than adults. Swimming is a water contact activity that contributes most transmission. Children swim often, especially during warmer months and invariably urinate while doing so, thus introducing continual supply directly into the water [13]. However, a clear understanding of the snail species, their local distribution and infection status is therefore essential for effective control of schistosomiasis [14].

Previous KAP studies have shown differential knowledge levels among communities. A systematic literature review on schistosomiasis revealed that awareness and knowledge was low mainly among rural communities [15]. Other studies conducted in Ethiopia, Ghana, Côte d'Ivoire, Tanzania, and Mozambique also showed low levels of knowledge of the diseases [16–18] while other studies conducted in Zimbabwe and Kenya showed higher level of knowledge [18]. The studies concluded that low levels of knowledge of the disease makes communities vulnerable and at risk of infection. Others [19, 20] explored the effect of gender on risk factors, more so that gender roles may dictate the frequency of water contacts among males and females, thereby pointing to exposure to risky behaviors. A study in Uganda found no differences among boys and girls in the frequency of water contact activities, while in adults, women had more water activities compared to men [20]. A meta-analysis revealed that males were more vulnerable to reinfection than females [19]. Other demographic characteristics showing strong association with reinfection included age [21].

In Botswana, the highest prevalence of *S. mansoni* infection (just over 80%) was recorded in 1983, 6 years after the exceptionally high in-flows of 1977. This wet period was followed by the dry conditions of the 1980s, including several dry spells where the Thamalakane River did not receive water for several consecutive months. This decline in flow was accompanied by a decline in the prevalence of *S. mansoni* infection, with low values recorded in 1989 (8.4%) and 1991 (6.7%) that match the declines in flow seen 6 years prior to each of these dates [4]. Comparison of flows with the prevalence of schistosome infection in young school children

indicates that there is an inverse relationship between the rise of flow and fall of schistosome transmission, Appleton and colleagues further noted that there was little evidence of the active transmission of *S. hematobium* in the delta over the last few decades [4]. None of the communities in Ngamiland district mapped by Doumenge in 1987 had prevalence of *S. haematobium* infection exceeding 6.8% [22], although a prevalence of 32% at Pandamatenga was recorded [23]. This dominance of *S. mansoni* over *S. haematobium* is not confined to the Okavango Delta.

In Botswana, the termination of the National Schistosomiasis Control Program in 1993 contributed to its neglect. However, an outbreak of schistosomiasis in 2017 at one of the primary schools in the northeastern part of the country, resulted in 42 positive cases, an indication that the disease exists. Therefore, this study sought to determine the prevalence of urogenital and intestinal schistosomiasis among school age children 6 to 13 years in selected communities in the Okavango Delta.

## 2. Methods

### 2.1 Study design

We used the previous minimum estimates of the prevalence found by Appleton and colleagues [4] to estimate the minimum required sample size for a cross-sectional study design. With the prevalence of 20%, probability of type 1 error of 0.05 and the margin of error of 0.05, the estimated minimum sample size was 246. Assuming a non-response of 20%, the minimum sample size required is 308. However, a descriptive cross-sectional survey was administered to a total of 1,611 school age children 6–13 years who were randomly selected from the school registers in 10 primary schools from a population.

### 2.2 Study area and population

The study villages Kauxwi, Shakawe, Etsha 6 and Maun were purposively sampled based on their location across the three sections of the Okavango Delta: upper, middle and the distal part.

Kauxwi is situated on the north-eastern part of the panhandle and Shakawe on the south-western side. The two villages are about 10 km apart across the Okavango River. Etsha 6 is situated in the middle section, while Maun, the major urban centre, is in the south-eastern side, the distal section of the Okavango Delta, straddling the Thamalakane River. The three different sections of the Okavango Delta, on which the study sites are located, present varied hydrological conditions and flooding regimes that may pose a risk to bilharzia.

Secondly, schools within the four study sites were also purposively sampled based on their location to the river. Kauxwi Primary School was the only primary school in the village, while Shakawe village had two primary schools, Shakawe and Kathiana which were all selected because of their proximity to the river. Etsha 6 Primary School was selected as the only primary school in Etsha 6. In Maun, six primary schools were selected for being the closest to the Thamalakane River. These were Mathiba, Botswelelo, Thamalakane, Moremi, Letsholathebe and Thito Primary.

Lastly, respondents were sampled using stratified random sampling procedure based on grade or standard level to ensure representativeness [24]. School registers were sought from the school management, and where a grade had more than one class, the names were combined into one list to make one sampling frame. From each grade level, 40% of the children were randomly sampled using computer generated random numbers to take part in the school survey. Sample sizes per school are indicated in Table 1.

**Table 1. Total enrolment of school age children in the 4 study sites and actual sample selected.**

| Village | Primary Schools | Total Enrolment | Total Sample | Actual Sample |
|---------|-----------------|-----------------|--------------|---------------|
| Maun | Letsholathebe | 678 | 272 | 206 |
| | Moremi | 607 | 242 | 112 |
| | Thito | 211 | 85 | 55 |
| | Botswelelo | 438 | 175 | 115 |
| | Thamalakane | 363 | 142 | 110 |
| | Mathiba | 908 | 363 | 275 |
| Etsha 6 | Etsha 6 | 635 | 254 | 185 |
| Shakawe | Kathiana | 702 | 283 | 111 |
| | Shakawe | 721 | 291 | 262 |
| Kauxwi | Kauxwi | 465 | 187 | 180 |
| **Total** | | **5728** | **2294** | **1611** |

## 2.3 Ethical considerations

Ethical approval was obtained from the University of Botswana Institutional Review Board and the Health Research Development Committee of the Ministry of Health and Wellness (HPDME: 13/18/1). Permission was also sought from the Botswana Ministry of Basic Education, Ngamiland Regional Education Office. We obtained written consent from parents and legal guardians and oral assent from children. The purpose of the study and the interview process were described, and the interviewers ensured that the children were comfortable. The assent form was read to the children, and they were requested to complete the form if they agreed to participate in the study, which was completely voluntary. They were assured of confidentiality and that they had the right not to answer any or some of the questions that made them feel uncomfortable and that they had the right to withdraw from the study at any time without penalty. Data were analyzed anonymously. Children who were found positive for *S. mansoni* and *S. hematobium* were treated according to the WHO clinical guidelines and Ministry of Health and Wellness.

## 2.4 Data collection

**2.4.1 Parasitology.** A total of 1603 urine and 1404 stool samples were collected from school age children in the selected schools to determine the prevalence and intensity of schistosomiasis. Macroscopic examination of urine for color, odor and blood was conducted for all urine samples. 10ml of urine was processed by filtration and examined for parasite ova at x40 magnification for schistosoma ova. To increase the sensitivity of detecting parasite ova, urine filtration and centrifugation methods were used to analyze urine samples concurrently [25–27]. Positive diagnosis of *S. haematobium* was based on the detection of one terminal spined schistosoma ovum or more.

Kato-Katz and Formalin-Ether methods were used for the examination of stool samples. The technique was used to detect the presence of *S. mansoni* and other parasite ova in single stool specimens. All Stool samples were examined macroscopically for viscosity, color, odor, consistency, and the presence of worms. Kato-Katz and Formalin-Ether methods were run concurrently on each stool sample collected. The slides were examined under a microscope at x40 objective lens by experienced laboratory technicians and a medical laboratory scientist who counted the number of *S. mansoni* eggs and STH in the stool. Infection intensity was determined by multiplying the number of ova detected on Kato-katz by 24 to determine the number of eggs per gram(epg) of stool [28–30].

## 2.5 Survey questionnaire

Primary data were collected using a structured survey questionnaire, capturing the socio-demographics of the participant, household water and sanitation facilities, family livelihood, hobbies and recreational activities, information sources, knowledge, and practices regarding bilharzia. The survey instruments were reviewed for face and content validity by a panel of experts within the research team. All questionnaires were translated into Setswana and back translated into English.

**2.5.1 Data management and statistical analysis.** Data quality assurance was ensured through training of research assistants on basic interviewing skills involving children, data capturing, coding, and processing of parasitology samples. Researchers did random spot-checks to review completed questionnaires. Data was entered into a password protected computer.

Quantitative data were analyzed using SPSS version 25. Means and standard deviations were used to summarize continuous variables while frequencies and proportions were used for categorical data such as knowledge, water, sanitation, and hygiene (WASH) facilities, recreational and parasitology results for the overall sample. To test for differences between categorical variables, we used the Fisher's Exact test. For continuous ones, we used t-tests and one-way between-group ANOVA to test for significant differences among groups. Logistic regression was conducted to examine the effect of a set of selected predictor variables on likelihood of swimming in stagnant water, being a risky behavior. Assumptions for t-test, ANOVA and logistic regression were tested and found tenable. Prevalence of infection was estimated by dividing the number of individuals who tested positive with the total number of children examined. The results were expressed as odds ratio (*OR*) with their 95% CI and statistical significance set at $p < 0.05$.

## 3. Results

**3.1**. A total of (n = 1611) school age children 6 to 13 years participated in the study, mean age 9.73 years (SD 2.06), females (54%) and males (46%).

## 3.2 Parasitology results

The results indicate that Shakawe and Kauxwi Primary Schools had each 34 (12% and 21%) cases of *S. haematobium* while Kathiana and Estha 6 had 9 and 6 (3%) cases respectively. Cases of *S. mansoni* were very few ranging between 0.4–1% as indicated in Table 2. Although other soil transmitted helminths were not the focus of this study; Ascaris was highest (n = 20) in Shakawe Primary School than in schools.

Parasitology results from Maun primary schools reported few cases of *S. haemotobium* and *S. mansoni* and other soil transmitted helminths (see Table 3). Although other soil transmitted helminths were not the focus of this study; Ascaris and Taenia were identified, and about 50% (n = 14) was in Mathiba Primary School.

**Table 2. Parasitology results in the upper panhandle and middle sections of the Okavango Delta.**

| School | Urine (N) | *SS. hematobium* | Stool (N) | *S. mansoni* |
|---|---|---|---|---|
| Shakawe | 276 | 34 | 260 | 1 |
| Kathiana | 286 | 9 | 272 | 1 |
| Kauwxi | 169 | 34 | 150 | 2 |
| Etsha 6 | 178 | 6 | 165 | 1 |
| | 909 | 83 | 847 | 5 |

**Table 3. Parasitology results of primary schools in Maun the distal part of the Okavango Delta.**

| School | Urine (N) | SS. hematobium | Stool (N) | S. mansoni |
|---|---|---|---|---|
| Thamalakane | 43 | 0 | 43 | 0 |
| Mathiba | 259 | 0 | 151 | 0 |
| Botswelelo | 146 | 2 | 146 | 0 |
| Thito | 45 | 0 | 45 | 0 |
| Moremi | 108 | 0 | 87 | 0 |
| Letsholathebe | 93 | 1 | 85 | 2 |
| | 694 | 3 | 557 | 2 |

## 3.3 Prevalence and Intensity of schistosomiasis

The overall prevalence of S. haematobium was 8.7%, and for S. mansoni 0.64% respectively. Other soil transmitted helminthes, *Ascaris*10.9% and hookworm 1.06%. Intensity for S. haematobium was generally light at 97.6% (n = 40), and heavy intensity 2.4% (n = 1) (see Table 4).

The prevalence and intensity of S.mansoni was very low at 0.64% (see Table 5) below, this was also reflected in the parasitology results (Tables 3 and 4).

## 3.4 Knowledge of schistosomiasis

An independent-samples t-test revealed significant differences in age amongst females and males, with mean age for males (M = 9.95, SD = 2.13) a little bit higher than that of females (Table 6). Most learners 58% (n = 917) had not heard of bilharzia. Although more girls (n = 362, 53.7%) were aware of bilharzia than boys (n = 312, 47%), statistically there was no significant association between gender and awareness of bilharzia ($\chi^2$ (1, n = 1590) = .15, p = 0.70, phi = 0.01) An independent-samples t-test conducted to compare knowledge scores for i) bilharzia risk activities, ii) signs and symptoms and iii) overall combined knowledge scores by gender revealed no significance difference in scores for males and females (Table 6). The overall knowledge mean score is 4.71, SD = 2.68, signifying low knowledge of schistosomiasis.

Table 6 shows that according to the Fisher's Exact test, only age was different among the gender groups, p = 0.00. One-way between-groups ANOVA was conducted to examine the effect of grade on knowledge scores of bilharzia, measured on total scores each pupil got correct on bilharzia risk activities, signs and symptoms. Learners were in their pre-existing grade level, being Standard 1, 2, 3, 4, 5, and 6. There were significant statistical differences among the six grade level groups: $F_{(5, 637)}$ = 16.92, p = 0 .00, with a large effect size (eta squared) of 0.11. Post hoc comparisons using Tukey HSD revealed significant differences between Grade 1 and all other Grades; Grade 2 significantly differed with three Grades (1, 3, and 6); Grade 3 with Grades 1 and 2, Grade 4 significantly differed with Grade 1 only; Grade 5 also differed with Grade 1 only while Grade 6 differed significantly with Grades 1 and 2. Assessment of mean scores indicates that Grade 1 scored lowest among all grades (M = 2.24; SD = 2.87) (Table 7).

**Table 4. Prevalence and Intensity of *S. hematobium*.**

| S. Hematobium | Infection | N (%) |
|---|---|---|
| Not infected | | 430 (91.3) |
| Infected | | 41 (8.7) |
| Light intensity | Intensity | 40 (97.6) |
| Heavy intensity | | 1 (2.4) |

**Table 5. Prevalence and intensity of S. mansoni.**

| S. mansoni | Infection | N (%) |
|---|---|---|
| Not infected | | 469 (99.4) |
| Infected | | 3 (0.64) |
| No intensity | Intensity | 2 (66.7) |
| Light intensity | | 1 (33.3) |

**Table 6. Association between selected variable by gender.**

| Variable | N | Females (n%) | Males (n%) | F | P | Effect |
|---|---|---|---|---|---|---|
| Age | 1607 | 9.59 (1.99) | 9.89 (2.13) | 2.95 | 0.003 | |
| Awareness of Schistosomiasis | 1590 | 53.7 (362) | 46.3 (312) | 0.15 | 0.70 | 0.01 |
| Knowledge of Bilharzia risk factors | 772 | 2.86 (1.6) | 2.81 (1.46) | -0.44 | 0.66 | |
| Knowledge of bilharzia Signs and symptoms | 649 | 1.99 (1.50) | 1.93 (1.53) | -0.44 | 0.66 | |
| Knowledge composite | 648 | 4.76 (2.68) | 4.63 (2.69) | -0.62 | 0.54 | |

Superscript[1] means p-value for Fishers' exact test has been reported. Superscript means that the mean, standard deviation, and p-value for t-test have been reported.

**Table 7. Learners' knowledge mean scores by grade level.**

| Grade | N | M | SD | SE | 95% CI | |
|---|---|---|---|---|---|---|
| | | | | | LB | UB |
| 1 | 63 | 2.24 | 2.87 | 0.36 | 1.52 | 2.96 |
| 2 | 72 | 3.92 | 3.36 | 0.40 | 3.13 | 4.71 |
| 3 | 109 | 5.07 | 2.44 | 0.23 | 4.61 | 5.54 |
| 4 | 117 | 4.98 | 2.49 | 0.23 | 4.53 | 5.44 |
| 5 | 126 | 4.90 | 2.48 | 0.22 | 4.47 | 5.34 |
| 6 | 156 | 5.46 | 2.06 | 0.17 | 5.13 | 5.78 |
| **Total** | **643** | **4.71** | **2.69** | **0.11** | **4.50** | **4.92** |

Furthermore, Table 3 shows that the mean scores increased from Grade 1 (being the lowest) to Grade 3, then decreased and almost levelled between Grade 4 and 5 and ultimately increased for Grade 6, which has the highest mean scores.

An independence-sample t-test was conducted to compare bilharzia knowledge scores for learners who had had a family member who suffered bilharzia and those who did not. There was a significance difference in mean knowledge scores for those who have had a family member who previously suffered from bilharzia (M = 5.98, SD = 1.90) and those who never had a family member infected with bilharzia (M = 4.56, SD = 2.78; t (80.76) = 4.94, p = 0.00). The effect size was very small (eta squared = 0.04), implying that only 4% of the variance in knowledge scores were explained by whether one had a family member infected with bilharzia or not. Mean knowledge scores for learners who had a family member previously infected with bilharzia was higher than those who did not. The mean difference was 1.42, 95% CI [0.85, 1.99].

Logistic regression was conducted to examine the effect of a set of selected predictor variables on the likelihood of swimming in stagnant water, being a risky behavior. The explanatory variables were grade level, gender, knowledge of bilharzia and whether one had had a family member who suffered from bilharzia while the response variable was whether one has swum in stagnant water or not. The overall model was statistically significant, $\chi^2$ (8, N = 626) = 36.01,

**Table 8. Logistic regression predicting likelihood of engaging in a risky behavior.**

| | B | SE | Wald | Df | P | Odds Ratio | 95% CI for Odds Ratio | |
|---|---|---|---|---|---|---|---|---|
| | | | | | | | Lower | Upper |
| Grade | | | 3.80 | 5 | 0.58 | | | |
| Grade 1 | 0.04 | 0.42 | 0.01 | 1 | 0.92 | 1.04 | 0.46 | 2.38 |
| Grade 2 | -0.49 | 0.41 | 1.48 | 1 | 0.22 | 0.61 | 0.28 | 1.35 |
| Grade 3 | 0.01 | 0.39 | 0.00 | 1 | 0.98 | 1.01 | 0.47 | 2.16 |
| Grade 4 | -0.28 | 0.39 | 0.50 | 1 | 0.48 | 0.76 | 0.35 | 1.63 |
| Grade 5 | -0.14 | 0.38 | 0.13 | 1 | 0.72 | 0.87 | 0.42 | 1.83 |
| Gender | 0.23 | 0.19 | 1.51 | 1 | 0.22 | 1.26 | 0.87 | 1.80 |
| Knowledge of disease | 0.16 | 0.04 | 19.70 | 1 | 0.00 | 1.19 | 1.10 | 1.29 |
| Family member infected | -0.76 | 0.31 | 5.94 | 1 | 0.02 | 0.47 | 0.26 | 0.86 |
| Constant | -1.07 | 0.47 | 5.26 | 1 | 0.02 | 0.35 | | |

$p < .001$. The model was able to correctly predict 72.7% of the cases and this was a small improvement from the null model which only classified 71.9% cases correctly. Knowledge of the disease and previous cases of infection within the family made a unique statistically significant contribution to the model, while grade level and gender did not (Table 8).

The results show the odds ratio of 1.19 for knowledge of disease (bilharzia), indicating that for every additional unit of knowledge of bilharzia, the odds were 1.19 times higher that respondents would report swimming in stagnant water, controlling for other factors. This shows that among the learners, the higher the knowledge of bilharzia, the higher the likelihood of engaging in risky behaviors, such as swimming in stagnant water. For learners whose family members had suffered from bilharzia, the odds of swimming in stagnant water were 0.47 lower than those who did not have a family member infected with bilharzia. This implies that respondents who had family members infected with bilharzia were less likely to report swimming in stagnant water, thereby engaging in risky behavior.

### 3.5 Availability of sanitation and water facilities

Regarding availability of water and sanitation facilities, 55% used pit latrines, 23% the bush and 21.3% flush toilets. Those who used pit latrines (n = 879) and those who used the bush were at higher risk of contracting infection n = 366 (22.7%), OR = 0.7 (CI; 0.0–0.8), and [343 (21.3%), OR = 0.5(CI;0.0–0.9)]. The findings also indicate that about 49% accessed water from the river, while 26.9% and 1.4% accessed it from reserve tanks *(Jojo)* and boreholes respectively. Most (80.6%) reported that swimming in infected water, drinking dirty water (75%), walking barefooted in dirty water (75%) and washing clothes in dirty water (53%) could transmit infection.

In addition, 43.1% reported that they swim; of these 70.3% swam in the river or stream and that they swam over weekends and school holidays. However, only a few (6.2%, n = 77) reported previous infection, while (6.7%) had a family member diagnosed with schistosomiasis in the past.

### 4. Discussion

Since the termination of the national program in 1993, schistosomiasis has received little attention. The termination of the program has over the years hampered the goals set by the World Health Assembly Resolution [31] to be achieved in Botswana. The WHA Resolution advocated for the control of schistosomiasis morbidity in highly endemic areas and urged countries to

attain a target of regular treatment of "at least 75% and up to 100% of all school-aged children at risk of morbidity by 2010" [31]. The schistosomiasis program was terminated when it became clear that prevalence of the parasite remained below 10% and intestinal schistosomiasis was no longer a public health problem in the district [4]; thus, surveillance and monitoring were halted. Some studies have argued that schistosomiasis and other soil transmitted helminths are neglected diseases which are perceived to have a very low burden of disease index. Globally, more than a third of the world's population are currently infected and children suffer from profound physical deficits such as anemia, malnutrition, stunted growth, and cognitive delays [32].

This study revealed a knowledge deficit, about 58% of children had never heard of bilharzia even though they lived in communities where the disease was previously endemic. In fact, with the termination of the program, the health training institutions and the Ministry of Education excluded NTDs in their curricula. The knowledge deficit was observed among the school age children at different grade levels. The lower grades learners' knowledge of schistosomiasis was lowest and increased with grades. This is because health education was more emphasized at higher grades than lower ones. The finding is not surprising as similar results were reported in a Swaziland study, where knowledge discrepancies regarding schistosomiasis among primary school children were revealed [33]. The renewed attention on NTDs by WHO in 2017 is timely for countries like Botswana to curb morbidity that may result from the endemic and chronic nature of schistosomiasis [34]. Another interesting finding in this study is that learners who had a family member who previously suffered schistosomiasis had higher knowledge than those who did not. This is in line with other studies which showed that learners from areas (e.g., schools) with high prevalence of schistosomiasis showed higher knowledge of the disease and took preventive measures against reinfection [35]. Another study conducted among children in Yemen also found that children from families with history of schistosomiasis infections had higher knowledge than those who did not [36]. This may be the case when family members are infected with schistosomiasis, they learn about the disease.

The results also showed knowledge and history of disease within family as determinants of schistosomiasis preventive behaviors. It is surprising that learners with higher knowledge of schistosomiasis are likely to engage in risky behaviors compared to those with lower knowledge of the disease. Though surprising, previous studies in health behavior have shown that knowledge on its own does not lead to desired health behavior [37–39]. The finding indicates that knowledge alone may not predict behavior as there are multiple factors that are needed to promote pro-health behaviors. Environment, peers, cues to action and situational factors can also play a role. It is therefore necessary that billboards are placed next to stagnant water pools as cues and public swimming facilities. Okavango sub-district does not have recreational facilities such as public swimming pools where adolescents may relax. Providing knowledge without alternatives may not be effective on its own.

In this study, an overall prevalence of *S.hematobium* and *S.mansoni* was 8.7% and 0.64% respectively. In addition, the results demonstrated that the intensity of schistosomiasis was generally light at 97.6% and heavy intensity at 2.4%; this calls for continued monitoring of morbidity and that treatment is required.

It is possible that the prevalence of schistosomiasis was higher in the upper and mid sections of the Okavango Delta because, during data collection the Okavango River had more water in Kauxwi, Shakawe and Etsha 6 than in Maun, the distal part. Children in the upper and mid Okavango were observed swimming in the river, as such, an association between swimming in infected water and the risk of infection was established. A lot of children were observed swimming in the two communities which seemed to predispose them to infection [40]. They attributed the hyperendemicity of the disease in the two communities to be intensified by water

contact activities such as bathing, fishing, and swimming in cercariae infested streams, rivers, and ponds.

In Maun, the results might have been influenced by the drought spell the country experienced, more specifically during data collection, the Thamalakane River water level was very low, and this is the river which runs through this urban village. This could explain the low infection transmission and low prevalence rates. In addition, the study findings demonstrate that while *S. Mansoni* might be on the decline, *S. haematobium* is on the rise, which makes it a public health concern. Mass drug administration (MDA) with praziquantel was conducted in all schools where children tested positive, but this should be repeated to make sure treatment is effective. The current WHO recommendation is that in areas with a prevalence over 10% but less than 50% MDA should be conducted once every two years to be able to break and sustain the control of schistosomiasis transmission. We recommend that all school age children 6–13 years in the Okavango Delta should receive praziquantel once every two years to sustain the transmission of schistosomiasis. Mass drug administration should be carried out together with other schistosomiasis control strategies such as snail control and health education to reach elimination goals. In 2017, the World Health Assembly adopted resolution [34], calling member states to develop or adapt national vector control strategies and operational plans aimed at reducing by at least 40% the incidence of vector borne diseases, including schistosomiasis by 2025. WHO reinforces snail control as part of its strategic approach to eliminate schistosomiasis as a public health problem.

Ngamiland district like other districts in Botswana, still has limited access to safe drinking water (running water and protected wells or rivers) and sanitation (flush toilets and pit latrines). In this study, almost half (49%) fetched water from the river. The study further revealed that the main source of portable water which are community standpipes were dry most of the time, further putting these communities at risk for WASH related diseases. This use of pit latrines and open defecation which is still practiced has the potential of contaminating water sources. To control and prevent transmission of schistosomiasis, public health education is critical and effective measures should be put in place to improve water supply, sanitation, and hygiene infrastructure. In addition, public health interventions that focus on behavioral change and WASH measures may enhance effects of MDA against schistosomiasis and STH [41].

### 4.1 Limitations of study

This study had some limitations, the single screening of urine and stool specimens. Most studies recommend collection of samples on two consecutive days from the children to optimize recovery of eggs and better estimate infection level and true prevalence in those communities [40, 42]. Secondly, in some areas the processing site of samples was too distant, and we are not sure that this could have in anyway affected urine specimen. Thirdly, the study was conducted when the Delta was very dry hence low prevalence rates. Lastly, the research team could not repeat the mass drug administration to children who had tested positive because of the Covid-19 movement restriction.

### 5. Conclusions

Our study indicated an overall prevalence of *SS. hematobium* and *S.mansoni* at 8.7% and 0.64% respectively. Intensity of *SS. hematobium* was generally light (97.6%) and heavy intensity (2.4%). The findings also revealed a knowledge deficit, about 58% of children had never heard of bilharzia even though they lived in communities where the disease was previously endemic. Learners who had a family member who previously suffered from schistosomiasis had higher

knowledge than those who did not. Interestingly, these learners were likely to engage in risky behaviors compared to those with lower knowledge of the disease. An integrated approach including health education, mass drug administration, surveillance, water, sanitation, and hygiene infrastructure should be prioritized for the prevention and control of schistosomiasis in the Botswana national NTD strategy. There is also an urgent need to undertake a large-scale study to determine schistosomiasis prevalence nationwide.

## Supporting information

**S1 File.**
(SAV)

## Acknowledgments

We thank the Ministry of Basic Education for availing the primary schools in Ngamiland; the Ministry of Health and Wellness for availing their laboratory facilities as well as provide praziquantel treatment; the Community Leadership in the study sites, the community, and parents for allowing their children to participate in the study. Our heartfelt gratitude to the study participants, our research assistants, and our postgraduate students. We would also like to extend our gratitude to Mr. Kesaobaka Molebatsi for his support with statistical data analysis.

## Author Contributions

**Conceptualization:** Nthabiseng A. Phaladze, Lebotse Molefi, Olekae T. Thakadu, Onalenna Tsima, Barbara N. Ngwenya, Tuduetso L. Molefi, Wananani B. Tshiamo.

**Data curation:** Nthabiseng A. Phaladze, Barbara N. Ngwenya.

**Formal analysis:** Nthabiseng A. Phaladze, Lebotse Molefi, Olekae T. Thakadu, Onalenna Tsima, Barbara N. Ngwenya, Tuduetso L. Molefi, Wananani B. Tshiamo.

**Funding acquisition:** Nthabiseng A. Phaladze, Olekae T. Thakadu, Barbara N. Ngwenya.

**Investigation:** Nthabiseng A. Phaladze, Lebotse Molefi, Olekae T. Thakadu, Onalenna Tsima, Barbara N. Ngwenya, Tuduetso L. Molefi.

**Methodology:** Nthabiseng A. Phaladze, Lebotse Molefi, Olekae T. Thakadu, Onalenna Tsima, Barbara N. Ngwenya, Tuduetso L. Molefi, Wananani B. Tshiamo.

**Project administration:** Nthabiseng A. Phaladze.

**Resources:** Nthabiseng A. Phaladze, Olekae T. Thakadu, Barbara N. Ngwenya.

**Software:** Nthabiseng A. Phaladze, Olekae T. Thakadu.

**Supervision:** Nthabiseng A. Phaladze, Lebotse Molefi, Olekae T. Thakadu, Onalenna Tsima, Barbara N. Ngwenya, Tuduetso L. Molefi.

**Validation:** Nthabiseng A. Phaladze, Lebotse Molefi, Olekae T. Thakadu, Onalenna Tsima, Barbara N. Ngwenya, Tuduetso L. Molefi, Wananani B. Tshiamo.

**Visualization:** Nthabiseng A. Phaladze, Lebotse Molefi, Olekae T. Thakadu, Onalenna Tsima, Barbara N. Ngwenya, Tuduetso L. Molefi, Wananani B. Tshiamo.

**Writing – original draft:** Nthabiseng A. Phaladze, Lebotse Molefi.

**Writing – review & editing:** Nthabiseng A. Phaladze, Lebotse Molefi, Olekae T. Thakadu, Onalenna Tsima, Barbara N. Ngwenya, Tuduetso L. Molefi, Wananani B. Tshiamo.

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
