## [Decision Letter · Decision Letter 0]

7 Dec 2022

PONE-D-22-26018The prevalence of schistosomiasis among primary school aged children (6-13 years) in the Okavango Delta in Botswana.PLOS ONE

Dear Dr. Phaladze,

Thank you for submitting your manuscript to PLOS ONE. After careful consideration, we feel that it has merit but does not fully meet PLOS ONE’s publication criteria as it currently stands. Therefore, we invite you to submit a revised version of the manuscript that addresses the points raised during the review process.

We look forward to receiving your revised manuscript.

Kind regards,

Clement Ameh Yaro, Ph.D

Academic Editor

PLOS ONE

Journal Requirements:

"The authors have read the journal’s policy and have the following competing interests: LM is a paid employee of Quality Anchor Consultants PTY.LTD, but was not affiliated with the company at the time this study was conducted. There are no patents, products in development or marketed products associated with this research to declare. This does not alter our adherence to PLOS ONE policies on sharing data and materials.”

"This research was commissioned by the National Institute for Health Research (NIHR) Global Health Research programme (16/136/33) using UK aid from the UK Government. The views expressed in this publication are those of the authors and not necessarily those of the NIHR or the Department of Health and Social Care."

"This study was funded by the National Institute of Health Research (Grant Number: 16/136/33 NIHR). The funders had no role in study design, data collection and analysis, decision to publish, or preparation of the manuscript.”

4. We note that Figures 1 and 2 in your submission contain [map/satellite] images which may be copyrighted. All PLOS content is published under the Creative Commons Attribution License (CC BY 4.0), which means that the manuscript, images, and Supporting Information files will be freely available online, and any third party is permitted to access, download, copy, distribute, and use these materials in any way, even commercially, with proper attribution. For these reasons, we cannot publish previously copyrighted maps or satellite images created using proprietary data, such as Google software (Google Maps, Street View, and Earth). For more information, see our copyright guidelines: http://journals.plos.org/plosone/s/licenses-and-copyright.

a. You may seek permission from the original copyright holder of Figures 1 and 2 to publish the content specifically under the CC BY 4.0 license.  

Reviewers' comments:

Reviewer's Responses to Questions

**Comments to the Author**

1. Is the manuscript technically sound, and do the data support the conclusions?

Reviewer #1: Partly

Reviewer #2: Yes

2. Has the statistical analysis been performed appropriately and rigorously? 

Reviewer #1: Yes

Reviewer #2: Yes

3. Have the authors made all data underlying the findings in their manuscript fully available?

Reviewer #1: Yes

Reviewer #2: Yes

4. Is the manuscript presented in an intelligible fashion and written in standard English?

Reviewer #1: No

Reviewer #2: Yes

5. Review Comments to the Author

Reviewer #1: Reviewer’s Comments________________________________________

Title: The prevalence of schistosomiasis among primary school aged children (6-13 years) in the Okavango Delta in Botswana

Manuscript Number: PONE-D-22-26018

Overall comments

The paper is of interest to schistosomiasis control program of Botswana as it provides information on re-emerging of the disease after the termination of the Botswana national schistosomiasis control program in 1993. Had the snail survey been conducted along the parasitological survey, the study would have more informed the schistosomiasis control program of the country.

The primary objective/purpose of this study was to determine the prevalence and intensity of schistosomiasis among the study participants. The title also reflects this to some extent. The paper should also be structured accordingly in all sections, i.e., presenting prevalence and intensity first, followed by participants’ knowledge of schistosomiasis, and so on.

Although the paper reports on important information it is not well structured and written. Hence, it needs substantial revision to bring it to the acceptable level.

Specific comments

Title:

It is suggested that the term “schistosomiasis” in the title be replaced with “intestinal and urogenital schistosomiasis”. Additionally, the authors need to replace the phrase “...primary school aged children (6-13 years)...” in the title and throughout in the document be replaced with “...school-age children...”

Abstract:

Page 1,Background: provide purpose/objective of the study

Page 1,Methods: more details required for methods used for stool and urine examination, as well as for statistical analysis of the data.

Page 3, lines 46- 48: Recommendation should be preceded by summary of the main findings.

Page 7: It would be appropriate to create the heading “Study area and population” with which the method section begins.

Page 7, lines 118 - 121: It is not necessary to outline specific objectives in an article as it is not a proposal. It is usual to provide purpose /objective statement only.

Page 9, lines 164 & 165: For individuals who have not reached the legal age of consent, you only need consent from parents or legal guardians. Individuals who have not reached the legal age of consent are not expected to complete the assent form but only give oral assent.

Page 10, lines 183 & 184: The statement that reads “Total Urine (n=1603) and stool (n=1404) samples collected from school aged children from all the selected schools to determine the prevalence and intensity of schistosomiasis.” lacks clarity and needs to be re-phrased to read “A total of 1603 urine and 1404 stool samples were collected from school age children in the selected schools to determine the prevalence and intensity of schistosomiasis.” This is just one example, otherwise there are many other such vague statements for which the paper needs considerable editorial revision.

Page 11, lines 185 & 186: For examination of S. haematobium infection, WHO recommends urine filtration method. It is not clear why the authors used additional centrifugation. The authors need to clarify this. The authors also need to re-write the urine and stool examination methods and cite appropriate references.

Page 14, Table 2: vertical lines in the table should be removed.

Page 19, Table 5:

The figures in column “S. haematobium” add up to 83, and not to 156.

It is not clear why “Ascaris/Taenia” is in the table.

Show STH identified in the footnote.

Page 20, Table 6:

It is not clear why “Ascaris/Taenia” is in the table.

Show STH identified in the footnote.

Page 21, Table 7: The number of S. haematobium infected cases were 86(83 in Table 5 & 3 in Table 6). On the other hand, the authors have written the number of S. haematobium infected cases as 41. The discrepancy should be reconciled.

Page 22, Table 8: The number infected with S. mansoni was 3 while the intensity of infection was presented for 1. Here correction is needed.

Page 24, line 412: “S. Haematobium” on this line and in other section should be corrected as “S. haematobium”

Page 27, lines 464 & 465: The statement “Our data provide baseline information regarding infection status of Schistosoma and STH among school aged children living along the Okavango Delta.” is a significant statement and should be replaced with a summary of the findings.

Reviewer #2: The manuscript provides interesting data at a time when the activities on Neglected Tropical Diseases is aiming for a global elimination. Such situation and existing conditions require to be highlighted. The data provides vividly that there are missing gaps in the whole approach in the road to elimination as depicted by the situiation in the Okavango region. This may not be an isolated existing condition but could be prevailing in other regions in the sub Saharan region. I have identified some sections that require proper presentation in conformity to the generally required for publications.

1. I suggest that the figures given in all the tables should have an integer for example where it is given as .40 % it should be 0.40%. This suggestion is for all the presnetation in the manuscript.

2. Line 66 the reference need to be written in a proper way instead of the listed names. Usually it is the only first author as (Hajissa et al., 2018)

3. Line 85 should read as : Other studies conducted in Ethiopia...

4. line 87: Conducted in Zimbabwe and Kenya....

5. line 183: Total urine (n=1603 and stool ...

6. line 187: ova indicated a positive diagnosis of S. heamatobium ...

7. The methods should not be listed as points but contained in a discriptive way.

8. Lines 349 and 359 the Genus and species names: S. heamatobium and S. mansoni..

9. Line 390 grammar: who had a family member who previously suffered ....

10. Remove information between 409 - 422 as it is repetition of lines 394 -396.

11. Line 412: S. haematobium

12. Lines 433 and 436: reconstruction of the sentence. sustain control of schistosomiasis transmission...

13. Line 461. COVID-19 restriction

References section require proper presentation.

Ref 506, 509: The author is the same Appleton CC or Appleton C?

ref 527: Presentation of the author is not clear if, B Doumbo,OK

Care should be taken in the use of reference manager where the format has changed fromn Ref 593 - 626.

6. PLOS authors have the option to publish the peer review history of their article (what does this mean?). If published, this will include your full peer review and any attached files.

Reviewer #1: **Yes: **Berhanu Erko

Reviewer #2: **Yes: **Professor Takafira Mduluza

---

## [Author Response · Author response to Decision Letter 0]

16 Mar 2023

We are pleased to resubmit a manuscript PONE-D-22-26018 The prevalence of urogenital and intestinal schistosomiasis among school age children (6-13 years) in the Okavango Delta in Botswana by Nthabiseng A. Phaladze (Principal investigator), and co-authors Lebotse Molefi, Olekae T. Thakadu, Onalenna Tsima, Barbara N. Ngwenya, Tuduetso L. Molefi and Wananani B. Tshiamo for consideration to be published in PLOS One. 

We wish to take this opportunity to heartily thank the reviewers for their valuable time and useful contribution to this work. We appreciate the thorough inputs they have given which will no doubt help improve our manuscript. The following paragraphs contain point-by-point responses to the reviewers’ and editorial board comments:

A. Response on Journal Requirements: 

1. Compliance of the Manuscript to PLOS ONE’s style requirements: We are happy to submit the revised manuscript in line with all the requirements of the journal. 

2. Competing Interests Statement: I declare that the authors have no conflict of interest. Mr . Lebotse Molefi is a paid employee of Quality Anchor Consultants PTY. LTD, but was not affiliated with the study at the time of this study was conducted. There are no patents, products in development or marketed products associated with this research to declare. This does not alter our adherence to PLOS ONE policies on sharing data and materials.”

3. Funder: Funding information has been removed from the Acknowledgments section. We would like to maintain the Funding Statement as it reads "This study was funded by the National Institute of Health Research (Grant Number: 16/136/33 NIHR). The funders had no role in study design, data collection and analysis, decision to publish, or preparation of the manuscript.” Any additional information provided contrary to the original statement should be deleted.

4. Copyright of Figures 1 & 2: have been deleted as advised since it was difficult to get permission from the copyright holder. 

B. Comments to the Author:

1. Is the manuscript technically sound, and do the data support the conclusions?

We note that Reviewers agree on this aspect. We have also noted comments made by Reviewer #1. 

2. Has the statistical analysis been performed appropriately and rigorously?

We note that Reviewers agree on the statistical analysis.

3. Have the authors made all data underlying the findings in their manuscript fully available?

The PLOS Data policy requires authors to make all data underlying the findings described in their manuscript fully available without restriction, with rare exception (please refer to the Data Availability Statement in the manuscript PDF file). The data should be provided as part of the manuscript or its supporting information or deposited to a public repository. For example, in addition to summary statistics, the data points behind means, medians and variance measures should be available. If there are restrictions on publicly sharing data—e.g. participant privacy or use of data from a third party—those must be specified.

We also note that Reviewers agree on this aspect.

4. Is the manuscript presented in an intelligible fashion and written in standard English?

PLOS ONE does not copyedit accepted manuscripts, so the language in 

submitted articles must be clear, correct, and unambiguous. Any 

typographical or grammatical errors should be corrected at revision, so please

 note any specific errors here.

We do note the Reviewers divergent views on this question; but do take comments made by Reviewer #1 seriously and have attended to typographical and grammatical errors.

5. Review Comments to the Author:

a. Reviewer #1

1. Title: The suggestion to change the title to “The prevalence of urogenital and intestinal schistosomiasis among school age children (6-13 years) in the Okavango Delta in Botswana” has been adopted. 

The phrase “primary school aged children (6-13 years)” in the title and throughout in the document has been replaced with “school-age children in the title and throughout the text.

2. Abstract: Page 1, Background: line 1, the study purpose has been provided “This study sought to investigate prevalence and intensity of schistosomiasis among school age children 6-13 years in selected communities in the Okavango Delta”. 

- Methods: More details of methods used for stool and urine examination, as well as for statistical analysis of data have been provided.

- Recommendation preceded by main findings as per advice.

3. Page 7, Specific objectives have been removed, and the study purpose maintained.

4. Page 7, Heading created “Study area and population”.

5. Ethical considerations: statement on assent by children corrected on page 9, lines 156-157.

6. The statement that reads “Total Urine (n=1603) and stool (n=1404) samples collected from school aged children from all the selected schools to determine the prevalence and intensity of schistosomiasis.” was re-phrased as suggested to read “A total of 1603 urine and 1404 stool samples were collected from school age children in the selected schools to determine the prevalence and intensity of schistosomiasis.” More details on methods used to analyze urine and stool samples are provided on Page 10.

7. Comment: For examination of S. haematobium infection, WHO recommends urine filtration method. It is not clear why the authors used additional centrifugation. The authors need to clarify this.

- Clarification: The majority of intestinal, urinary and blood parasites

 can be detected microscopically in unstained or stained preparations, 

either directly or following concentration by centrifugation. (Medical 

Laboratory Manual for Tropical Countries Volume 1 page 178 (second 

edition by Monica Cheesbrough (ELBS-English Language Book 

Society/Tropical Health Technology/ 1987).

 To increase the sensitivity of detecting parasite ova, urine filtration

 and centrifugation methods were used to analyze urine samples 

concurrently. Positive diagnosis of S. haematobium was based on the 

detection of one terminal spined schistosoma ovum or more. 

8. The urine and stool examination methods have been re-written and appropriate references cited.

9. Comments on Tables 2;5;6;7 &8 have been addressed and statistical corrections done. The issue of other soil transmitted helminths -Ascaris, Taenia have been removed from the tables. PLOS ONE does not allow footnotes. We added a sentence in the text to acknowledge that even though other STHs were not the focus, during urine and stool examination, they were identified.

10. The statement “Our data provide baseline information regarding infection status of Schistosoma and STH among school aged children living along the Okavango Delta.” The statement has been replaced with a summary of the findings.

b. Reviewer #2:

1. Integer inserted in all Figures given in all the tables and throughout the 

 manuscript.

2. Line 66: Referencing of Hajissa et al., 2018 corrected.

3. Lines 85; 87; 183 & 187 corrected as per advice.

4. Methods of examination for urine and stool samples written in a 

 narrative form; bullets removed. 

5. Lines 349, 359 and 412: Genus and species names corrected as per

 advice.

6. Repetition of information removed, and grammar corrected.

7. Lines 433 and 436: Reconstruction of the sentence corrected as 

 suggested.

8. Line 461: Covid-19 restriction accepted.

9. References Section corrected.

We hope these responses meet your expectations and awaiting your decision.

Sincerely,

Nthabiseng A. Phaladze PhD.

Professor of Nursing

University of Botswana School of Nursing

Email: phaladze@ub.ac.bw

Alternate email: nphaladze@yahoo.co.uk

---

## [Decision Letter · Decision Letter 1]

7 May 2023

The prevalence of urogenital and intestinal schistosomiasis among school age children (6-13 years) in the Okavango Delta in Botswana.

PONE-D-22-26018R1

Dear Dr. Phaladze,

We’re pleased to inform you that your manuscript has been judged scientifically suitable for publication and will be formally accepted for publication once it meets all outstanding technical requirements.

Kind regards,

Clement Ameh Yaro, Ph.D

Academic Editor

PLOS ONE

Additional Editor Comments (optional):

Reviewers' comments:

Reviewer's Responses to Questions

**Comments to the Author**

1. If the authors have adequately addressed your comments raised in a previous round of review and you feel that this manuscript is now acceptable for publication, you may indicate that here to bypass the “Comments to the Author” section, enter your conflict of interest statement in the “Confidential to Editor” section, and submit your "Accept" recommendation.

Reviewer #1: All comments have been addressed

2. Is the manuscript technically sound, and do the data support the conclusions?

Reviewer #1: Yes

3. Has the statistical analysis been performed appropriately and rigorously? 

Reviewer #1: Yes

4. Have the authors made all data underlying the findings in their manuscript fully available?

Reviewer #1: Yes

5. Is the manuscript presented in an intelligible fashion and written in standard English?

Reviewer #1: Yes

6. Review Comments to the Author

Reviewer #1: (No Response)

7. PLOS authors have the option to publish the peer review history of their article (what does this mean?). If published, this will include your full peer review and any attached files.

Reviewer #1: **Yes: **Berhanu Erko

<quillbot-extension-portal></quillbot-extension-portal>

---

## [Editor Report · Acceptance letter]

16 May 2023

PONE-D-22-26018R1 

The prevalence of urogenital and intestinal *schistosomiasis* among school age children (6-13 years) in the Okavango Delta in Botswana. 

Dear Dr. Phaladze:

I'm pleased to inform you that your manuscript has been deemed suitable for publication in PLOS ONE. Congratulations! Your manuscript is now with our production department. 

Kind regards, 

on behalf of

Dr. Clement Ameh Yaro 

Academic Editor

PLOS ONE